# Positive Performance Feedback and Innovation Search: New Ideas for Sustainable Business Development

**Yongbo Sun and Zichen Qiu ***

Business School, Beijing Technology and Business University, Fucheng Road, Beijing 100048, China; sunyb@th.btbu.edu.cn
* Correspondence: qiuzichen0102@163.com; Tel.: +86-132-63225859

**Abstract:** Although the literature suggests that firms tend to adopt "conservative" behavior in the face of positive performance feedback, there are also studies that take the opposite view on the impact of positive performance feedback. Based on the behavior theory of the firm and regulatory focus theory, this study explored the impact of positive performance feedback on innovation search behavior and the boundary effect of CEO regulatory focus to gain insight into the mechanisms of innovation search behavior and to promote innovation for sustainable development. Based on data from 230 biopharmaceutical companies in China, the analysis found that: (1) positive performance feedback had a significant positive effect on depth search behavior and a significant negative effect on breadth search behavior; (2) CEOs' promotion focus had a negative moderating effect on the relationship between positive performance feedback and depth search behavior and a positive moderating effect on the relationship between positive performance feedback and breadth search behavior; and (3) CEOs' prevention focus positively moderated the relationship between positive performance feedback in relation to depth search behavior. This study extends the behavior theory of the firm and reveals the mechanism of the differential impact of positive performance feedback on innovation search behavior, which has implications for the study of which innovation search practices should be conducted by high-performing firms to promote sustainable development.

**Keywords:** positive performance feedback; innovation search; regulatory focus; sustainable development

## 1. Introduction

In the context of open innovation, enterprises' thirst for innovation resources is increasing day by day. Faced with limited resources, innovation search behavior, as a way to collect and learn information such as knowledge and technology to increase the novelty of resources, is an important way for enterprises to break through the status quo in order to obtain a wider range of resources and improve the innovation of enterprises [1,2]. However, in the existing literature, there are many studies on the impact of innovation search behavior on innovation performance, but the existing studies are scarce and paradoxical regarding the influencing factors of antecedent variables of innovation. In today's rapidly evolving society with faster product iterations and changing consumer needs, the market competition environment is becoming increasingly fierce, which leads to higher standards for enterprises to carry out innovation activities. More and more enterprise managers find that by relying on their own resources, they cannot adapt to the changing market environment and cannot obtain sustainable innovation advantages [3]. Enterprises need to actively conduct innovative search behaviors to obtain resources according to their own state, and innovative resources need not only include internal resources but also external resources, which are open [4]. In this context, the open innovation behavior of enterprises is promoted through innovation search behavior, and sustainable competitive advantages are obtained to ensure the sustainable development of enterprises [5].

The relationship between a firm's performance feedback and innovation search behavior has been one of the main focuses of academic research in this field. Most studies

have concluded that when there is positive performance feedback, firms are in a state of satisfaction and lack the motivation and willingness to proactively search, which may lead them to avoid risk-taking behaviors, and when there is negative performance feedback, firms are eager to change the status quo and increase their innovation efforts in problem searching to escape the situation [6–8]. In contrast, other studies have shown that when positive performance feedback occurs [9,10], it indicates slack resources of the firm, and slack resources support more exploration and trial and error, making the firm more inclusive of failure, which in turn promotes innovation search behavior [11]. This view elaborates that good performance feedback increases firms' innovation search behavior. It is easy to see that there is a paradox in the established studies. To resolve the paradox of previous studies, this paper reveals the impact of positive performance feedback on innovation search behavior based on the behavior theory of the firm and regulatory focus theory to gain insight into the mechanism of innovation search behavior and promote sustainable development of firms.

The regulatory focus theory suggests that the psychological characteristics of the CEO play a key role in choosing strategies, such as overconfidence and overly conservative behavior [12]. Recent studies have indicated that regulatory focus plays a key role in CEO goal orientation [13,14]. This attribute reflects the motivation of individuals to dare to innovate (promotion focus) and conservative mindset (prevention focus). Research indicates that regulatory focus has a critical impact on CEOs' strategic choices, especially for those that are highly uncertain [15]. Although CEOs are actively involved in the strategic choices of their firms [16] and the trade-offs between risk-taking and conservatism, studies examining the impact of CEOs' regulatory focus on firms' strategic choices remain scarce, implying that the impact of CEO regulatory focus on strategy may be underestimated. The innovation search behavior studied in this work is divided into depth search and breadth search behavior, both of which require CEOs to trade-off between innovativeness and riskiness [17–19]. From this, we propose that CEO regulatory focus may influence firms' search behavior choices in the face of the same positive performance feedback.

This study is expected to make the following contributions: first, to expand the empirical study of the behavior theory of the firm by selecting data from biopharmaceutical companies with high R&D investment to argue for an influential relationship between positive performance feedback and innovation search behavior, to explore corporate innovation search mechanisms for sustainable corporate development, and to explain the paradoxes of previous studies; second, to extend the literature on CEO regulatory focus by proposing that CEO regulation focus affects the relationship between positive performance feedback and innovation search behavior; third, to help governments and capital markets analyze the factors influencing innovation search behavior in high performing firms by providing a research perspective.

This paper argues that the sustainable development of enterprises needs a sustainable competitive advantage, the sustainable competitive advantage comes from the innovation ability of enterprises, and the prerequisite of innovation capability enhancement needs to be ensured by choosing the most appropriate innovation search strategy at the right stage. Therefore, this argument was taken as the main clue in this study to analyze innovation search behavior as a means to explore new ideas for enterprises to achieve sustainable development.

## 2. Literature Review and Theoretical Hypothesis

Performance feedback can be divided into positive performance feedback and negative performance feedback based on the comparison with past performance, and early research suggested that positive performance feedback leads firms to tend to be risk-averse and reduce risk-taking behavior [20]. Recent studies on the impact of positive performance feedback on firms' innovation search behavior have exhibited mixed findings in the literature: Based on social pressure theory [21], when firms experience positive performance, they revise their future performance goals upwards and the pressure to achieve their next goals

rises [9,10]. Firms actively engage in risk-taking behavior in order to maintain their position and relative competitive advantage in the market. Analyzed in terms of redundancy-driven search, [11] positive performance feedback implies the existence of slack resources available to the firm that can provide a buffer for firms to engage in risk-taking behavior, making the organization more tolerant of failure and promoting risk-taking behavior. Higher-order theories suggest that positive performance feedback increases managers' ego levels, making them believe that they can control risky exploratory behaviors and are more inclined to choose risky behaviors as they focus more on the gains behind them and ignore the possible losses when performing risky behaviors [9]. Different studies based on different theoretical perspectives have inconsistencies regarding the role of positive performance feedback in the innovation search behavior, with mixed findings from multiple theoretical foundations. The behavior theory of the firm states that when an organization is close to the expected level, it chooses to seek risks below the level and avoid risks above the level, which further affects the organization's reform or innovation performance improvement [22]. Based on the behavior theory of the firm and Katila and Ahuja's research, firms may engage in breadth search and depth search behavior when faced with performance feedback. Breadth search behavior is the process of organizational exploratory learning, which refers to the integration of a broader range of technologies into a new technological track, opening up new areas, with high difficulty, high resource requirements, and high-risk, high-cost innovation attributes; in contrast to breadth search behavior, depth search behavior emphasizes the understanding and deepening of existing knowledge and technologies, with low difficulty, low resource requirements, low risk, and low cost [23]. The two different types of innovation search have different requirements for enterprise resources and risk tolerance, and enterprises should carefully choose the appropriate innovation search method according to their own ability and development needs.

### 2.1. Positive Performance Feedback and Innovation Search

This study links positive performance feedback to innovation search behavior and explores the ambidextrous nature of innovation search behavior. Existing research on the performance expectation feedback largely assumes that innovation search behavior is fixed [24]. However, the market is changing, and the resource environment faced by enterprises and the speed of research and development of biopharmaceuticals cannot remain unchanged, so the innovation search behavior should not be static but dynamic. This study helps companies analyze the impact of positive performance feedback on two dimensions of innovation search behavior and how to choose a suitable method in the face of dynamic changes in the environment so as to help companies improve the utilization of resources, enhance innovation, and achieve sustainable development [25,26].

Positive performance feedback indicates that the firm is growing well during the current operating period, is "relatively rich", and is motivated to innovate. CEOs often carry out innovative behaviors driven by innovation motivation [21], such as innovation search behavior. However, this paper argues that higher-performing firms may respond differently to different innovation search behaviors.

First, the manageability of risk in the familiar path: Firms that experience positive performance feedback typically have more idle resources (e.g., capital, equipment, and human resources) [27]. Although the success of operations during this period leads to the existence of some idle resources, firms aiming to gain a sustained competitive advantage to achieve sustainable growth need to seek new breakthrough solutions and perform increasingly well relative to competitors, which requires innovation search behavior [28]. The abundance of idle resources compensates for the cost of time and uncertainty consumed by firms conducting innovation search behavior and also encourages CEOs to experiment and explore [29]. The available idle resources are limited. In the process of running a business, making changes to the way the organization strategically allocates resources not only raises the CEO's own hiring risk [30] but also risks professional reputation acquired from the previous business success, which poses a double threat of loss for the CEO and

increases the fearfulness and psychological cost of innovation search behavior for the CEO. In order to reduce the employment risk and maintain the acquired professional reputation, the CEO prefers the plan with low decision risk [31]. With the motivation of "seeking stability", put idle resources into existing fields, conduct depth research and development, reduce risks to a controllable range, and depth search behavior has become a better behavioral choice.

Second, it enhances the CEO's decision-making confidence on a familiar path. Positive performance feedback proves the effectiveness of the current strategy of the enterprise, and better performance also improves the confidence of CEO in the early strategic choice, confirming that the CEO is correct in the utilization of resources and the grasp of development direction [9]. This also leads CEOs to believe that control over the use of internal corporate resources is effective and that they have some experience and competence in the development and exploration of external resources [32]. In the current era of open innovation, CEOs are also aware that internal resources alone are not sufficient and that they need to draw more innovative knowledge from external sources to expand the resource base of the firm in order to improve the firm's innovation performance. Innovation search behavior has been identified as the preferred strategy for CEOs to facilitate innovation [33]. To some extent, CEOs are influenced by their past technological development trajectories because past successful experiences make CEOs confident in their existing strengths. Therefore, risk and experience are more likely to focus CEOs' searches on familiar paths during innovation search behavior. CEOs are thus more likely to focus on a familiar path. Depth search behavior using existing channels can reduce the possibility of errors and increase the predictability of search [34,35]. In addition, CEOs in the successful state have lower motivation to search for information [36] because early successes have made the CEO more confident. Accordingly, CEOs tend to conduct depth searches to find profits. Thus, we hypothesize:

**Hypothesis 1 (H1).** *Positive performance feedback negatively affects the breadth search behavior of enterprise innovation.*

**Hypothesis 2 (H2).** *Positive performance feedback positively affects the depth search behavior of enterprise innovation.*

### 2.2. Moderating Role of Regulatory Focus: CEO Regulatory Focus

According to the regulatory focus theory, two different motivational systems influence how individuals view risk taking and security: promotion focus and prevention focus. The promotion focus is the individual's motivation to pay attention to progress and growth and tend to pursue risks and opportunities; the prevention focus is the motivation of individuals to avoid losses and ensure safety and tend to avoid risks and pursue stability [30,37,38].

Established studies have indicated that decisions between involving organizational gain capture and loss avoidance can be predicted by the regulatory focus of the CEO [38]. CEOs with a high promotion focus tend to seek new resources and challenges, while CEOs with a high defensive focus tend to seek security and conservatism [37,38].

### 2.2.1. CEO Promotion Focus

The biopharmaceutical enterprise studied in this work is an enterprise that relies on continuous R&D and innovation [39]. The biopharmaceutical industry has become high innovation ability and large resource consumption [40]. With the requirements of high innovation ability, innovation search behavior is indispensable, and different regulatory focus of CEOs may have different influences on innovation search preferences because of their different inclinations.

First, promotion-focused CEOs have an aggressive risk-taking spirit [17,18]. Promotion-focused CEOs have a high propensity for risky exploration by seeking more new knowledge and resources to supplement the missing parts of the business [41]. Second, promotion-focused CEOs have a strong desire for self-actualization needs and a goal orientation of "maximizing acquisition" because they place a high value on their own growth. People

with a high promotional focus prefer to maximize their achievements, which is related to the promotion focus on achieving "maximum goals" [42], pursuing their own growth and aspirations, and preferring to use the resources at their disposal to maximize their own growth and thus achieve their overall career goals [34]. Although this trial-and-error-based resource exploration activity may have great uncertainty, cost, and risk, the successful development of new fields is a huge contribution to biopharmaceutical companies, as well as to the growth of the CEO [43]. Conversely, the presence of positive performance feedback can encourage focused CEOs to pursue broader domains of exploration, at the expense of exploiting existing domains. Positive performance feedback provides good external conditions for CEOs to explore and take risks, and the presence of disposable resources and corporate strengths is more likely to motivate promotion CEOs to take risks and "maximize acquisition" orientation. CEOs driven by a promotional focus are more likely to break new ground for the organization and help the business gain more revenue when facing certain risks.

Therefore, when the CEO promotes focus at a high level, the degree of motivation to pursue innovation, risk-taking, and to obtain maximum progress increases, and the motivation to ensure minimal risk and stability in decision-making in familiar paths diminishes. Hence:

**Hypothesis 3a (H3a).** *CEO promotion focus positively moderates the effect of positive performance feedback on innovation search breadth.*

**Hypothesis 3b (H3b).** *CEO promotion focus negatively moderates the effect of positive performance feedback on innovation search depth.*

2.2.2. CEO Prevention Focus

First, prevention-focused CEOs have a stronger tendency to be risk-averse. Second, prevention-focused people usually pursue "minimum goals", tend to be stable [44], prefer more limited options, and value the enforcement of rules and regulations [45], directing individuals to focus on safety goals. They are more sensitive to negative information, value their reputation and external image, avoid failures and potential losses as much as possible, and prefer to adopt avoidance-based strategies in their behavioral strategies to obtain performance and protection [46]. In the process of running a business, the CEO takes the minimum goal to achieve profitability [19]. When CEOs with prevention focus traits conduct innovative searches, they apply their resources to familiar areas to the greatest extent possible. Prevention-focused CEOs rely more on familiar knowledge in their innovation search choices and base their exploration and innovation activities on existing empirical paths [47]. Focusing on established or easily accessible technologies and resources, they apply their expertise to familiar areas of activity to increase the efficiency of resource use and speed of development [48]. Furthermore, due to concerns about risk and cost, defensive-focused CEOs are more likely to choose familiar, successfully practiced paths for development and exploration to reduce the risk and uncertainty of search costs [49]. CEOs are more likely to choose familiar, successfully practiced paths to develop and explore in order to reduce riskiness and uncertain search costs [50].

Therefore, when the CEO prevention focus at a high level, the degree of motivation to pursue stability, security, and risk avoidance is enhanced, the motivation to ensure the lowest risk in familiar paths and stability in decision-making is enhanced, and the CEO is highly likely to choose actions with lower risk, familiar paths, and to pursue stability. Hence:

**Hypothesis 4a (H4a).** *CEO prevention focus negatively moderates the effect of positive performance feedback on innovation search breadth.*

**Hypothesis 4b (H4b).** *CEO prevention focus positively moderates the effect of positive performance feedback on innovation search depth.*

The theoretical model of this study is illustrated in Figure 1.

**Figure 1.** The theoretical model.

## 3. Methodology

### 3.1. Sample and Data Collection

This study used publicly available data on listed biopharmaceutical companies. The data were mainly obtained from CSMAR database as well as the China National Intellectual Property Administration. Firstly, applicant patents of Chinese inventions were qualified in the China National Intellectual Property Administration; all the patent numbers of each pharmaceutical drug to be investigated were searched; the obtained data were filtered, processed and organized by year; and the number of patents, the number of newly cited patents, and the number of repeatedly cited patents that each drug had in each year were recorded. The basic characteristics of enterprises in the pharmaceutical manufacturing industry, CEO characteristics, and related financial data were obtained from the CSMAR database center, and the obtained data were screened and processed. Through the identification of codes and enterprise names, the collected patent numbers were matched with the data obtained from the CSMAR database center, and for some missing or inconsistent data, comparison tests and original data obtained from the websites of Shanghai and Shenzhen stock exchanges were compared and supplemented as much as possible manually to finally determine the consistency of the data.

A total of 230 listed biopharmaceutical companies in Shanghai and Shenzhen with A-shares from 2013–2017 were used as the observation sample in this study. The listed biopharmaceutical companies were selected because, on the one hand, they can effectively avoid the bias of the study results caused by industry differences and, on the other hand, the industry is characterized by high R&D investment and high patent output, in which innovative knowledge is dispersed among large pharmaceutical companies, new biotechnology companies (NBFs), and academic organizations, and therefore it is characterized by very active collaborative activities [51]. In this study, the data were screened and processed to improve the accuracy and reliability of the study based on the following: exclusion of companies with extreme financial data and exclusion of companies dealing with special status such as *ST.

### 3.2. Variable Measurement

#### 3.2.1. Positive Performance Feedback

Referring to earlier scholarly studies, the return on assets (ROA) was chosen to measure firm performance [24,52,53]. $B_{u,t-1}$ was used to represent the social performance expectations of firm u in the past year, calculated as:

$$B_{u,t-1} = (1 - \alpha_1) \ M_{u,t-2} + \alpha_1 B_{u,t-2}$$

$\alpha$ represents the weight and takes a range of values (0, 1], again drawing from [24], and $\alpha_1$ is taken as 0.4.

Therefore, the social performance expectation $B_{u,t-1}$ of firm u at period $t-1$ is the weighted sum of the median social performance expectation $M_{u,t-2}$ of firm u at period $t-2$ and the social performance expectation $B_{u,t-2}$ at period $t-2$. The social performance feedback $(P_{u,t-1} - B_{u,t-1})$ of firm u in period $t-1$ is the difference between the actual performance $P_{u,t-1}$ of firm u at period $t-1$ and the social performance expectation $B_{u,t-1}$ of firm u at period $t-1$. If $P_{u,t-1} - B_{u,t-1} > 0$, the actual performance of firm u in period $t-1$ is considered to be higher than the social performance expectation, then $I_1 = 1$, otherwise $I_1 = 0$. Combined with the definition of $I_1$, $I_1$ and $P_{u,t-1} - B_{u,t-1}$ can be multiplied to obtain the following positive variables with censored tails: $I_1 * (P_{u,t-1} - B_{u,t-1}) > 0$, which is higher than the social performance expectation, is the degree of positive performance feedback. The larger the value, the greater the difference in actual performance above social expectations, expressed in positive performance feedback (psfb).

### 3.2.2. Innovation Search

Referring to Reference [51], innovation search breadth (*ISB*) is measured by assessing the proportion of patent citations of the focal firm in a given year for which the patent citation is not present in the patent citation list for the past five years. For example, if a firm A files five patents in a given year t and each patent cites five patents, the total number of citations is $5 \times 5 = 25$. If five of these 25 citations are new (i.e., not cited in the last five years), then the innovation search width for firm A in a given year t is $5 \div 25 = 0.2$. The value of innovation search width ranges from 0 to 1. The variable is calculated as follows:

$$ISB\text{it} = \frac{\text{Number of new patent citations it}}{\text{Total number of patent citations it}}$$

where i represents the focus enterprise, and t represents the year.

Innovation search depth (*ISD*) is measured by assessing the percentage of patent citations of the focal firm in a given year that are repeated in the list of patent citations for the past five years. For example, given a total number of citations of 25 (like the example above), if the citation of a patent occurred within five years prior to the given year t, with the number of citations per year being 2, 1, 2, 2, 2, 1, then the total number of repetitions is $2 + 1 + 2 + 2 + 2 + 2 + 1 = 10$, and the innovation search depth for the given year t is $10 \div 25 = 0.4$. The value of innovation search depth cannot be negative. The variable is calculated as follows:

$$ISD\text{it} = \frac{\sum\limits_{y=t-5}^{t-1} \text{Number of duplicate patent citations iy}}{\text{Total number of patent citations}}$$

where *i* represents the focus enterprise, *t* represents the year, and *y* represents one of the previous five years.

### 3.2.3. Measurement of CEO Regulatory Focus

Primarily drawing on the promotion and prevention lexicon developed and validated by [30], the future outlook sections of the sample companies' corporate annual reports were collected, and the keywords were analyzed for word frequency using Python software.

The lexicon used to test the level of promotional-focused traits was contextualized and borrowed from Reference [30] to derive words related to facilitation: accomplish, achieve, aspire, desire, progress, earn, expand, grow, gain, hope, ideal, improve, increase, momentum, optimism, speed, rapidity, and toward. Words related to prevention included: accurate, fear, anxiety, avoid, careful, conservative, defensive, responsibility, escape, failure, fear, loss, pain, prevention, protection, risk, security, threat, and vigilance.

### 3.2.4. Control variables

Innovation search behavior in a firm is a result of a combination of factors with reference to a wide range of studies [54], such as size of the firm, age, and number of patents. According to the existing literature, factors are classified into two levels:

(1) Firm level

1.  Firm age (age): The age of the firm, measured in years, reflects the experience of the organization. [55].
2.  Firm size (size): Studies have shown that larger firms tend to have higher investment in innovation. This study uses the total assets of the firm at the end of the period as a measure to indicate firm size, which is measured logarithmically [46].
3.  Firm ownership (own): State-owned and non-state-owned firms respond differently to innovation search behavior, with non-state-owned firms performing innovation search practices faster.
4.  Distance to bankruptcy (bank): Z = (1.2 × working capital divided by total assets) + (1.4 × retained earnings divided by total assets) + (3.3 × income before interest expense and taxes divided by total assets) + (0.6 × market value of equity divided by total liability) + (1.0 × sales divided by total assets) [52].
5.  R&D intensity (rd): R&D intensity can be used to control for its effect on innovation search behavior, calculated as RD input ÷ sales [56].
6.  Free cash flow (fcf): (money funds + trading financial assets + notes receivable) ÷ current liabilities.
7.  Market value of book (mtb): book value of equity divided by its market value.
8.  Patent stock (pant): the number of active patents owned in period t − 1 [7].
9.  Profitability (probi): Firm profitability is employed as a proxy for firm financial performance and is computed as the ratio of a firm's operating income to its sales in a given year-note. The unit is percentage [57].
10. Leverage (leve): measured as the ratio of total debt to total assets, a good representation of the financial structure of the company [55].
11. Growth rate (gro): sales growth rate, reflecting the good and bad business performance of the company during this period [55].

Because the stock of slack resources may function as a performance buffer, reducing the perceived need for search and change, we accounted for organizational slack [12,58]. Following prior studies, we included three measures of slack to control for the possibility of slack search [21].

12. Potential slack (ps): the inverse ratio of debt to equity. This type of slack reflects that firms with high financial leverage are less likely to obtain additional funds and thus have smaller potential resources.
13. Absorbed slack (as): the ratio of selling, general, and administrative expenses to sales. This type of slack represents the resources translated into material form, such as additional employees and staff members.
14. Unabsorbed slack (uas): the ratio of current assets to liabilities. Also referred to as financial slack, this type of slack gives decision-makers the greatest degree of freedom for resource allocation.

(2) Executive level

15. CEO age: current year − year of birth [59].
16. CEO gender: CEO gender is associated with risk-taking [55].

## 4. Data Analysis and Results

In this study, STATA 16.0 software was used to conduct descriptive statistics, correlation analysis, and hypothesis testing for the collected data. The data were processed as follows before the specific tests to ensure the consistency and validity of the model estimates: (1) to avoid the influence of outliers on the test results, the main continuous variables

were subjected to tail shrinking at the 1% level; (2) to avoid the influence of multicollinearity, the variables measured by the interaction term were centered; in addition, all explanatory and control variables entering the model were subjected to variance inflation factor (VIF) diagnostics, and the results showed that the VIF was about 2.21 (the mean value of VIF for both outcome variables was less than 2.21); thus, the problem of multicollinearity could be excluded; (3) to exclude the possible problems of heteroskedasticity, cross-sectional correlation, and serial correlation in the panel data, the study used Driscoll–Kraay standard deviation for estimation to obtain the standard error.

### 4.1. Descriptive Statistical Analysis

Tables 1 and 2 show the descriptive and correlation analysis between the variables, which showed correlation between the variables and preliminary support for the hypothesis.

**Table 1.** Descriptive statistical analysis results.

| Variable | Obs | Mean | Std. Dev. | Min | Max |
|---|---|---|---|---|---|
| 1. isb | 621 | 0.537 | 0.482 | 0 | 1 |
| 2. isd | 621 | 0.03 | 0.105 | 0 | 0.773 |
| 3. psfb | 621 | 0.018 | 0.033 | 0 | 0.14 |
| 4. pro | 621 | 0.011 | 0.008 | 0 | 0.032 |
| 5. pre | 621 | 0.004 | 0.004 | 0 | 0.019 |
| 6. age | 621 | 16.91 | 5.197 | 5 | 29 |
| 7. size | 621 | 9.565 | 0.459 | 8.637 | 10.792 |
| 8. own | 621 | 0.356 | 0.479 | 0 | 1 |
| 9. bank | 621 | 7.807 | 8.181 | 0.725 | 47.765 |
| 10. CEOgen | 621 | 0.958 | 0.2 | 0 | 1 |
| 11. CEOage | 621 | 50.316 | 6.987 | 33 | 68 |
| 12. rd | 621 | 0.038 | 0.037 | 0 | 0.218 |
| 13. fcf | 621 | 1.875 | 2.554 | 0.095 | 15.196 |
| 14. ps | 621 | 4.096 | 4.255 | 0.163 | 21.075 |
| 15. as | 621 | 0.356 | 0.166 | 0.088 | 0.775 |
| 16. uas | 621 | 3.812 | 4.039 | 0.572 | 24.085 |
| 17. mtb | 621 | 0.428 | 0.196 | 0 | 0.876 |
| 18. pant | 621 | 48.841 | 61.23 | 0 | 312 |
| 19. probi | 621 | 0.01 | 0 | 0.01 | 0.011 |
| 20. leve | 621 | 0.311 | 0.185 | 0.045 | 0.86 |
| 21. gro | 621 | 0.193 | 0.321 | −0.372 | 2.011 |

**Table 2.** Correlation analysis results.

| Variable | 1. isb | 2. isd | 3. psfb | 4. pro | 5. pre | 6. age | 7. size |
|---|---|---|---|---|---|---|---|
| 1. isb | | | | | | | |
| 2. isd | 0.1280 | | | | | | |
| 3. psfb | −0.0762 | 0.0230 | | | | | |
| 4. pro | −0.0665 | −0.0347 | −0.0105 | | | | |
| 5. pre | 0.0093 | −0.0336 | −0.0365 | 0.4074 | | | |
| 6. age | −0.0535 | −0.0593 | 0.1005 | 0.0693 | 0.0871 | | |
| 7. size | 0.1124 | −0.0103 | 0.0276 | 0.0324 | 0.0365 | 0.2546 | |
| 8. own | 0.2121 | 0.0241 | −0.0726 | −0.0162 | 0.0332 | 0.0563 | 0.2453 |
| 9. bank | −0.1844 | −0.0398 | 0.3066 | 0.0090 | −0.0114 | −0.0145 | −0.2404 |
| 10. CEOgen | 0.0572 | −0.0465 | 0.0209 | 0.0115 | 0.0364 | 0.0273 | 0.0299 |
| 11. CEOage | −0.0972 | −0.0027 | 0.0549 | 0.0404 | 0.0375 | 0.1774 | 0.0868 |
| 12. rd | −0.0866 | 0.0176 | 0.0683 | 0.0147 | 0.0092 | 0.0320 | −0.1451 |
| 13. fcf | −0.1074 | −0.0431 | 0.3477 | 0.0714 | 0.0869 | −0.0098 | −0.0378 |
| 14. ps | −0.1613 | −0.0494 | 0.3569 | 0.0497 | 0.0868 | −0.0323 | −0.0932 |
| 15. as | −0.0614 | 0.0453 | 0.0795 | 0.0345 | 0.0250 | 0.0322 | −0.0266 |
| 16. uas | −0.1414 | −0.0564 | 0.3513 | 0.0867 | 0.0977 | −0.0234 | −0.0972 |
| 17. mtb | 0.0582 | −0.0116 | −0.3353 | 0.0495 | 0.0332 | 0.0321 | 0.1287 |
| 18. pant | 0.2439 | 0.0446 | −0.0117 | −0.0170 | −0.0387 | 0.0480 | 0.2463 |
| 19. probi | −0.0351 | 0.0533 | 0.0129 | 0.0496 | −0.0257 | −0.0682 | −0.0548 |
| 20. leve | 0.1413 | 0.0332 | −0.3558 | −0.0478 | −0.0398 | 0.0516 | 0.1299 |
| 21. gro | 0.0055 | −0.0603 | 0.1062 | −0.0092 | 0.0719 | −0.0505 | −0.0672 |

**Table 2.** *Cont.*

| Variable | 8. own | 9. bank | 10. CEOgen | 11. CEOage | 12. rd | 13. fcf | 14. ps |
|---|---|---|---|---|---|---|---|
| 9. bank | −0.2401 | | | | | | |
| 10. CEOgen | 0.0210 | −0.0314 | | | | | |
| 11. CEOage | −0.0504 | 0.0729 | 0.0751 | | | | |
| 12. rd | −0.3510 | 0.2736 | 0.0273 | 0.1626 | | | |
| 13. fcf | 0.0256 | 0.2115 | 0.0112 | 0.0057 | 0.0254 | | |
| 14. ps | −0.0646 | 0.3759 | 0.0124 | 0.0236 | 0.0714 | 0.8008 | |
| 15. as | −0.1528 | 0.0859 | 0.0437 | 0.0709 | 0.2708 | −0.0556 | −0.0181 |
| 16. uas | −0.0237 | 0.3025 | 0.0274 | 0.0135 | 0.0405 | 0.9152 | 0.8778 |
| 17. mtb | 0.0837 | −0.2898 | −0.0037 | 0.0023 | −0.0774 | −0.2055 | −0.2137 |
| 18. pant | 0.0276 | −0.0985 | 0.0364 | −0.0402 | −0.0379 | −0.1855 | −0.2127 |
| 19. probi | −0.0649 | 0.0057 | 0.0569 | −0.0101 | 0.1043 | −0.0163 | −0.0183 |
| 20. leve | 0.1206 | −0.3292 | 0.0248 | −0.0604 | −0.0900 | −0.5585 | −0.7443 |
| 21. gro | −0.0837 | 0.0004 | −0.0488 | −0.0141 | 0.0369 | −0.0678 | −0.0353 |
| **Variable** | **15. as** | **16. uas** | **17. mtb** | **18. pant** | **19. probi** | **20. leve** | |
| 16. uas | −0.0908 | | | | | | |
| 17. mtb | −0.1818 | −0.2385 | | | | | |
| 18. pant | −0.0558 | −0.1958 | 0.1705 | | | | |
| 19. probi | 0.1817 | −0.0113 | −0.0301 | −0.0495 | | | |
| 20. leve | −0.1091 | −0.6215 | 0.2972 | 0.2206 | 0.0590 | | |
| 21. gro | 0.0580 | −0.0704 | 0.0045 | 0.0063 | −0.0366 | −0.0126 | |

### 4.2. Hypotheses Testing

Table 3 shows that positive performance feedback has a significant negative relationship with breadth search behavior (β = −1.614, *p* < 0.001) and a significant positive relationship with depth search behavior (β = 0.440, *p* < 0.01), and hypotheses H1 and H2 are supported by the data.

**Table 3.** The impact of positive performance feedback on innovation search behavior.

| | (1) isb | (2) isb | (3) isb | (4) isd | (5) isd | (6) isd |
|---|---|---|---|---|---|---|
| pro | −1.416 ** | −1.491 ** | −1.357 * | 0.082 | 0.103 | 0.061 |
| | (−2.63) | (−2.98) | (−2.58) | (0.47) | (0.54) | (0.29) |
| pre | 4.964 *** | 5.116 *** | 4.808 *** | 0.055 | 0.013 | 0.108 |
| | (4.52) | (5.18) | (4.43) | (0.21) | (0.05) | (0.43) |
| nsfb | −0.809 ** | −0.660 * | −0.665 ** | −0.271 ** | −0.311 ** | −0.310 ** |
| | (−2.92) | (−2.60) | (−2.67) | (−2.97) | (−3.19) | (−3.26) |
| nhfb | −0.254 | −0.358 | −0.375 | 0.135 | 0.163 | 0.168 + |
| | (−1.04) | (−1.28) | (−1.25) | (1.28) | (1.62) | (1.78) |
| age | −0.003 ** | −0.002 * | −0.002 * | −0.000 | −0.001 + | −0.001 + |
| | (−3.03) | (−2.23) | (−2.34) | (−1.08) | (−1.80) | (−1.76) |
| size | −0.005 | −0.003 | −0.003 | −0.021 *** | −0.021 *** | −0.021 *** |
| | (−0.18) | (−0.11) | (−0.12) | (−5.04) | (−5.34) | (−5.61) |
| own | 0.109 *** | 0.113 *** | 0.113 *** | 0.008 * | 0.007 * | 0.007 * |
| | (5.05) | (5.45) | (5.79) | (2.58) | (2.52) | (2.57) |
| bank | −0.003 *** | −0.003 ** | −0.003 ** | −0.001 + | −0.001 * | −0.001 + |
| | (−3.48) | (−3.21) | (−3.03) | (−1.71) | (−2.00) | (−1.97) |
| CEOgen | 0.162 *** | 0.169 *** | 0.163 *** | −0.058 ** | −0.060 ** | −0.058 ** |
| | (13.77) | (13.41) | (11.50) | (−3.16) | (−3.31) | (−3.01) |
| CEOage | −0.000 | −0.000 | −0.000 | 0.000 ** | 0.000 *** | 0.000 |
| | (−0.18) | (−0.35) | (−0.11) | (3.35) | (3.87) | (1.37) |
| rd | −0.404 ** | −0.461 ** | −0.430 ** | 0.073 | 0.089 | 0.079 |
| | (−2.82) | (−3.20) | (−3.11) | (0.91) | (1.17) | (0.96) |

**Table 3.** *Cont.*

| | (1) isb | (2) isb | (3) isb | (4) isd | (5) isd | (6) isd |
|---|---|---|---|---|---|---|
| fcf | 0.026 ** | 0.026 *** | 0.027 ** | −0.004 *** | −0.004 *** | −0.005 ** |
| | (3.23) | (3.44) | (3.28) | (−3.93) | (−3.97) | (−2.80) |
| ps | −0.021 * | −0.022 * | −0.022 * | −0.003 + | −0.002 | −0.002 + |
| | (−2.17) | (−2.29) | (−2.30) | (−1.71) | (−1.63) | (−1.80) |
| as | −0.212 ** | −0.264 *** | −0.269 *** | 0.100 *** | 0.115 *** | 0.116 *** |
| | (−3.08) | (−3.94) | (−3.85) | (7.06) | (9.95) | (8.37) |
| uas | 0.003 | 0.002 | 0.002 | 0.005 *** | 0.005 *** | 0.005 ** |
| | (0.25) | (0.19) | (0.20) | (3.52) | (3.52) | (3.17) |
| mtb | −0.285 *** | −0.317 *** | −0.320 *** | −0.018 | −0.010 | −0.009 |
| | (−5.18) | (−6.80) | (−7.04) | (−0.75) | (−0.36) | (−0.32) |
| pant | 0.001 * | 0.001 * | 0.001 * | −0.000 | −0.000 | −0.000 |
| | (2.36) | (2.25) | (2.23) | (−0.56) | (−0.13) | (−0.15) |
| probi | −48.980 | −14.076 | −12.125 | 5.743 | −3.761 | −4.401 |
| | (−1.42) | (−0.45) | (−0.37) | (0.27) | (−0.20) | (−0.23) |
| leve | −0.254 | −0.326 + | −0.318 | −0.010 *** | 0.009 | 0.007 |
| | (−1.33) | (−1.67) | (−1.62) | (−3.50) | (1.35) | (0.90) |
| gro | 0.079 * | 0.095 ** | 0.099 ** | −0.022 + | −0.026 * | −0.028 * |
| | (2.39) | (2.78) | (2.96) | (−1.94) | (−2.27) | (−2.37) |
| dum1 | 0.096 *** | 0.095 *** | 0.094 *** | 0.007 | 0.007 | 0.007 |
| | (3.42) | (3.55) | (3.59) | (1.06) | (1.06) | (1.09) |
| psfb | | −1.614 *** | −1.499 ** | | 0.440 * | 0.403 * |
| | | (−3.89) | (−3.17) | | (2.59) | (2.41) |
| psfb*pro | | | 68.477 * | | | −21.708 ** |
| | | | (2.09) | | | (−2.86) |
| psfb*pre | | | −99.814 * | | | 30.694 |
| | | | (−2.08) | | | (1.19) |
| _cons | 1.219 * | 0.919 * | 0.893 * | 0.202 | 0.284 | 0.292 |
| | (2.45) | (2.04) | (1.98) | (0.90) | (1.36) | (1.39) |
| r2_w | 0.135 | 0.139 | 0.141 | 0.042 | 0.047 | 0.049 |
| N | 621.000 | 621.000 | 621.000 | 621.000 | 621.000 | 621.000 |

Note: * $p < 0.05$, ** $p < 0.01$, *** $p < 0.001$.

The variables mentioned in Tables 1–3 are described in Table 4

**Table 4.** Variable descriptions for Tables 1–3.

| | | |
|---|---|---|
| Isb: innovation search breadth | Isd: innovation search depth | Psfb: positive social performance feedback |
| Pro: CEO promotion focus | Pre: CEO prevention focus | Age: firm age |
| Size: firm size | Own: firm ownership | Bank: distance to bankruptcy |
| CEOgen: CEO gender | CEOage: CEO age | Rd: R&D intensity |
| Fcf: free cash flow | Ps: potential slack | As: absorbed slack |
| Uas: unabsorbed slack | Mtb: market value of book | Pant: patent stock |
| Probi: profitability | Leve: leverage | Gro: growth rate |

Moderating effect of positive performance feedback to innovation breadth search behavior: After adding the interaction term of positive performance feedback to CEO promotion focus, the regression coefficient of the interaction term of positive performance feedback to CEO promotion focus was positively significant (β = 68.477, $p < 0.05$), and the results indicated that CEO promotion focus had a significant positive effect on breadth search behavior, suggesting that CEO promotion focus plays a positive moderating role in the relationship between positive performance feedback and innovation breadth search behavior. Hypothesis H3a was supported by the data. After adding the interaction term of positive performance feedback and CEO defense focus, the regression coefficient of the interaction term of positive performance feedback and CEO prevention focus was negatively significant (β = −99.814, $p < 0.05$), and the results indicated that CEO prevention focus had a significant negative effect on innovation breadth search behavior, suggesting that CEO prevention focus plays a negative moderating role in the relationship between positive performance feedback and innovation breadth search behavior. Hypothesis H4a was supported by the data.

Moderating effect of positive performance feedback on innovation depth search behavior: After adding the interaction term of positive performance feedback and CEO promotion focus, the regression coefficient of the interaction term of positive performance feedback and CEO promotion focus was negatively significant ($\beta = -21.708$, $p < 0.01$), and the results indicated that CEO promotion focus had a significant negative effect on depth search behavior, suggesting that CEO promotion focus plays a negative moderating role in the relationship between positive performance feedback and innovation depth search behavior. Hypothesis H3b was supported by the data. The regression coefficient of the interaction term between positive performance feedback and CEO prevention focus was not significant ($\beta = 30.694$, $p > 0.1$) after adding the interaction term of positive performance feedback and CEO prevention focus. Hypothesis H4b was not supported by the data.

Subdivision of H3 and H4: Taking the hypothesis of H4a and H4b as an example first, the processing of the collected data shows that H4a holds, but H4b is not significant. This means that although innovation search behavior can be divided into depth search and breadth search behavior, the effects of the same moderator on these two opposing relationships are not necessarily completely opposite. Therefore, in order to truly, reliably, and accurately express the regulatory relationship between variables, it was necessary to clearly divide H3 and H4 to form H3a, H3b, H4a, and H4b and express the regulatory relationship clearly and intuitively.

This study divided positive performance feedback, promotion focus, and prevention focus into high and low groups using plus or minus one standard deviation as the grouping criterion, and the moderating effects are plotted in Figures 2–4. As can be seen from Figure 2, the positive effect of positive performance feedback on breadth search behavior was stronger when the CEO was high promotion focus compared to the case of low promotion focus; as the effect of positive performance feedback increased, the positive effect of positive performance feedback on breadth search behavior was stronger compared to the low promotion focus CEOs, and the likelihood of breadth search behavior increased faster, i.e., the slope was greater, when compared to the high promotion focus CEOs. As can be seen from Figure 3, the negative effect of positive performance feedback on the breadth of innovation search behavior was stronger when the CEO had high prevention focus compared to the low prevention focus case; as the effect of positive performance feedback increased, the likelihood of conducting a breadth search decreased more quickly for high prevention focus CEOs compared to low defensive focus CEOs, i.e., the slope was greater. As can be seen from Figure 4, the negative effect of positive performance feedback on depth search behavior was stronger when the CEO had a high promotion focus compared to the case of a low promotion focus; as the effect of positive performance feedback increased, the likelihood of conducting innovation search behavior for breadth decreased more quickly for CEOs with a high promotion focus compared to CEOs with a low promotion focus, i.e., the slope was greater.

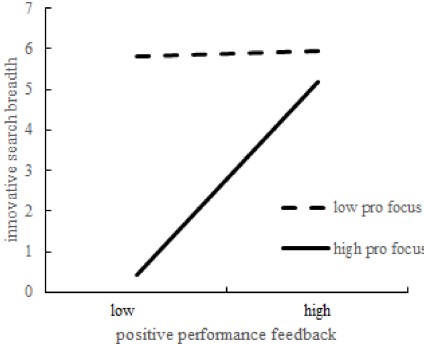

**Figure 2.** The moderating role of promotion focus in the role of positive performance feedback and breadth search behavior.

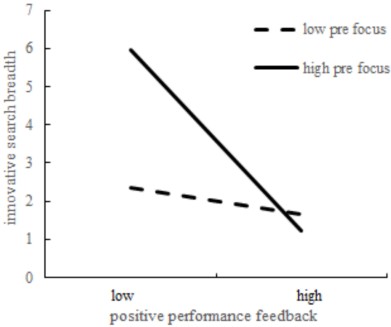

**Figure 3.** The moderating role of prevention focus in the role of positive performance feedback and breadth search behavior.

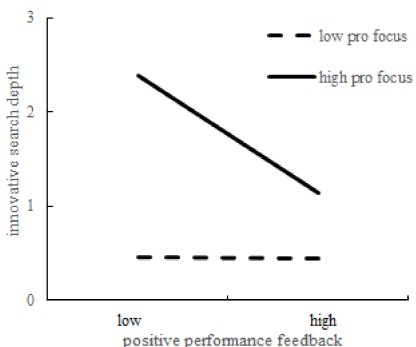

**Figure 4.** The moderating role of promotion focus in the role of positive performance feedback and depth search behavior.

## 5. Discussion and Conclusions

### 5.1. Discussion

The results of this study show that, first, positive performance feedback affects firms' innovation search behavior. Positive performance feedback positively influences innovation depth behaviors and negatively influences breadth search behaviors. For biopharmaceutical companies, positive performance feedback means that the organization is successful during the operation period and that there are redundant resources, which enhance the risk tolerance of the company, which in turn influences the company to conduct innovative searches. Biopharmaceutical enterprises are a high-tech organizations formed by the intersection of various disciplines, and at this stage, they are undergoing sustained and rapid development, characterized by high innovation, large capital investment, and high decision-making risks. With the entry of large multinational pharmaceutical groups and the rapid growth of large domestic biopharmaceutical groups, the pharmaceutical industry is also facing increasingly fierce competition. With positive performance feedback, facing the controllable risks on the familiar path and the CEO's decision-making confidence level make it easier for bio-enterprises to conduct innovative search behaviors on the familiar path, lock capital investment in familiar areas, promote innovation depth search behavior in the short term, minimize the decision-making risk while improving innovation behavior, enhance competitive advantages, promote sustainable development, and preventing large losses of funds, suppress breadth search behaviors in the short term, reduce risks, and ensure the normal operation of funds, which may also related to the fact that while the breadth search behavior consumes a lot of money, the acquired new domain knowledge cannot generate value in the short term.

Second, this study finds that CEO regulatory focus of a biopharmaceutical company moderates the effect of positive performance feedback on innovation search behavior. CEOs with high promotion focus have a high risk-taking spirit, show a strong sense of innovation, tend to boldly use the company's funds and resources to develop new innovations, develop new drugs or new technologies, and achieve "maximum achievement", thus

positively moderating the effect of positive performance feedback on innovation breadth search behavior and negatively moderating the effect of positive performance feedback on innovation depth search behavior. CEOs with a high degree of prevention awareness are more risk-averse and still maintain a stable attitude towards positive performance feedback, show a strong sense of safety despite operating at a level higher than half of the companies within the same industry competitors at a certain stage, take their own risk losses more seriously, have a high awareness of corporate capital and resource protection, and may not actively develop new fields of drug research and development and cross-field drug breakthroughs in the short term.

However, the results show that the effect of CEO prevention focus of a biopharmaceutical company on the moderating effect of positive performance feedback on this path of depth search behavior is not significant. The reason for the insignificant moderating effect on the depth search behavior, may be related to the length of the sample selection; when the firm has received positive performance feedback, the CEO, according to their own development plan and product characteristics, their development plan for the enterprise, the characteristics of the existing pharmaceutical technology, the difficulty of continuing to develop new drugs, and the length of the R&D cycle, may no longer carry out a large degree of development and innovation in the short term to accumulate funds and R&D strength for long-term strategic arrangements and development goals.

### 5.2. Theoretical Contributions

The behavior theory of the firm was extended to vary the role of positive performance feedback based on different innovation search contexts. This study shows that positive performance feedback has different effects on innovation search behavior, positively influencing firms to engage in depth search behavior and negatively influencing firms to engage in breadth search behavior. Existing studies on positive performance feedback and innovation search behavior tend to be based on different theoretical perspectives [60], including a problem-driven search perspective [11], a redundancy-driven search perspective [32], a prospect theory perspective for research [61], a social comparison theory perspective [62], a social pressure theory perspective and [9], a higher-order theory perspective. The different theoretical perspectives differ in their explanatory point of view, and the empirical evidence for performance feedback on innovation search behavior findings are mixed.

Second, it is demonstrated that CEOs with different focus have a facilitating or inhibiting effect on top-performing firms in specific search contexts. This finding also directly responds to the future research direction proposed by Li et al. (2018) and enriches the existing research. The few articles that have studied CEO focus only consider strategic variables such as level of diversification and capital structure [30] and rarely examine the impact of CEO regulatory focus on top-performing firms conducting innovation search practices. Therefore, this study provides good boundary conditions for the process of positive performance feedback in top-performing firms.

### 5.3. Managerial Implications

Enterprises need to know that they can achieve sustainable development by gaining a sustainable competitive advantage. One of the sources of a sustainable competitive advantage is the enterprise's own innovation ability, and the formation of innovation requires appropriate innovation search behavior. First, companies should pay attention to the relationship between positive performance feedback and innovation search behavior to understand the reasons for this phenomenon. This may be because, on the one hand, top-performing companies believe that the familiar path is low risk and can increase the CEO decision-making confidence. In the face of the fast-growing market, companies should fully understand the market and their own resource allocation and learn to reasonably break the rules while pioneering innovation to obtain a competitive advantage and a solid market position. To realize the necessity of enterprise innovation search behavior, in order to gain a competitive advantage in the fierce market competition, enterprises need to

formulate innovation strategies and actively cooperate with other related external fields when absorbing, transforming, and exploring their own resources. Secondly, CEOs in fast-growing companies should be aware of the role they play in regulating focus and can be associated with management teams that focus on prevention focus to fully understand the essential issues facing the development of the company and make decisions together to minimize the CEO's subjective focus bias. When it is clear that the mechanism of innovation is inherent in the company, it helps to promote innovation and achieve sustainable growth. In addition, even if the organization is in a stable environment, the CEO also needs to have the ability to face unexpected risks at any time and take decisive and correct measures to solve the crisis. Cultivating emergency response capabilities for crisis risks is also a capability that CEOs need to have. Especially for such highly R&D-focused and high investment biopharmaceutical companies, sudden risks may cause substantial losses. CEOs also need to have a sense of risk prediction to help companies prepare in advance, reduce possible losses in the future, preserve existing strengths, gain sustainable competitive advantages, and strengthen and promote sustainable development.

### 5.4. Limitations and Future Research

First, this study used a single dimension of depth and breadth for the division of innovation search behavior, but some scholars have already conducted multi-dimensional depth division in this field, which can be subdivided into studies in the future, such as local search versus remote search and exploratory search versus exploitative search. Sofka and Grimpe believe that the exploration of resource search behavior from the perspective of innovation depth and breadth cannot accurately reflect the differences in the attributes of knowledge elements provided by different types of resources [62], and such differences in knowledge elements may have an impact on innovative enterprises. Therefore, in the future, according to the characteristics of highly R&D-focused and high resource investment of biopharmaceutical enterprises, we can choose the search type that can provide the attributes of knowledge elements, explore the differentiated influence of different knowledge elements, carry out more targeted innovation, use limited resources to obtain sustainable competitive advantages, and promote the sustainable development of enterprises.

Second, due to the unavailability of secondary data from foreign companies, the analysis in this study was limited to Chinese biopharmaceutical companies. It will be possible to explore whether the innovation search choice is universal among other Chinese industries and top-performing foreign companies in the future. In the future, we can investigate whether other psychological characteristics of CEOs, values, educational background, etc. also have an impact on the choice of innovation behavior of biopharmaceutical companies because the development direction and speed of an enterprise are often related to its managers. The premise of the sustainable development of an enterprise is to obtain a sustainable competitive advantage. Whether an enterprise can sustain a competitive advantage is also related to the decision-making of managers, and the decision-making preferences of managers are also affected by their own psychological characteristics, values, and educational background.

Finally, for the research method of this study, QCA can be used for configurational analysis in the future. QCA advocates the use of a configurational perspective to explore the relationship between the elements of the problem, indicating that the elements that produce the results are related to each other, and the occurrence of the results is not caused by the only relationship established. In the future, the method of QCA can be used to explore the positive performance feedback of biopharmaceutical companies and other service manufacturing industries, the CEO's regulatory focus, and the combination mode of innovation search behavior to obtain sustainable competitive advantages and to achieve new ideas for sustainable development.

**Author Contributions:** Conceptualization, Y.S. and Z.Q.; methodology, Z.Q.; software, Z.Q.; validation, Z.Q.; formal analysis, Y.S. and Z.Q.; investigation, Y.S. and Z.Q.; resources, Y.S. and Z.Q.; data curation, Z.Q.; writing—original draft preparation, Z.Q.; writing—review and editing, Y.S.; visualization, Z.Q.; supervision, Y.S.; project administration, Y.S. and Z.Q.; funding acquisition, Y.S. All authors have read and agreed to the published version of the manuscript.

**Funding:** This research was supported by the National Social Science Fund of China (Grant number 18BGL083) and Beijing Natural Science Foundation of China (Grant number 9172007).

**Institutional Review Board Statement:** Not applicable.

**Informed Consent Statement:** Not applicable.

**Data Availability Statement:** Not applicable.

**Conflicts of Interest:** The authors declare no conflict of interest.

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
