# Peer review of "Positive Performance Feedback and Innovation Search: New Ideas for Sustainable Business Development"

_sustainability, doi:10.3390/su14042086_

Round 1

Reviewer 1 Report

Dear author(s),

I have gone through the paper titled: Positive performance feedback and innovation search: new ideas for sustainable business development. I find it well-written, with very interesting ideas and results. 

The theoretical part of the paper is well-systematized, and it contains adequate references and literature sources. I would like to ask you just to explain in more detail the innovation, as a theme, since you deal with the innovation search. I think you could just describe it in more detail because later you use a measurement of innovation, patents, and therefore, it would be nice to see a few rows on innovation. 

The methodology is also well written, and the data are appropriate.

The results are presented clearly and with sufficient explanations.

The recommendation is to make minor corrections and after that to accept this paper for publication.

Best regards.

Reviewer 2 Report

Basically, a well-written article in which it is definitely necessary to strengthen the link with sustainability. I miss the critical assessment of the literature analysis, which could definitely be improved.
The chapter on research limitations and future research opportunities is less developed. The author should make recommendations for future research.

Reviewer 3 Report

The paper addresses the problem of the relationship between performance feedback and innovation search. Undoubtedly, a great contribution of the authors is solving the paradox of previous studies and pointing to completely different relationships than in previous studies.

However, the authors took into account only biopharmaceutical companies with high R&D investment. The type of companies, the specifics of the market in which they operate and their efforts to constantly seek innovation may have had a significant impact on the result. Therefore, in my opinion, the subject of the research should be more strongly highlighted in the article and the conclusions should be limited to biopharmaceutical companies only.

The hypotheses H3a and H3b are not clear to me. Negative verification of hypothesis H3a means acceptance of the alternative hypothesis, i.e. H3b. It is therefore necessary to clarify the meaning of both hypotheses. The same applies to hypotheses H4a and H4b. This doubt should be clarified. Please also consider the hypotheses in Figure 1.

The article completely lacks the demonstration of the link between the research and sustainable business development. In fact, only in the title and very loosely in Abstract, Introduction and Theoretical Contributions the authors indicate this concept.

Furthermore, the following doubts arise regarding the estimation of variables:

1) How is the expected performance in period t-2 determined (line 216)?

2) In which formula is I1 taken into account (lines 227-228)?

3) What does "phfbo" mean? (line 230)?

4) How is the social performance expectation at period t-2 determined (line 233)?

5) In which formula is I2 taken into account (line 242)?

6) What do i and t mean in the formula for innovation search breadth (line 253)?

7) What do i, y and t mean in the formula for innovation search depth (line 262)?

8) There is no explanation of the variable symbols in Tables 1-3.

The authors write: "As can be seen from Figure 1, the positive effect of positive performance feedback on breadth search was stronger when the CEO was high facilitation focus compared to the case of low facilitation focus; as the effect of positive performance feedback increased, the positive effect of positive performance feedback on breadth search was stronger compared to the low promotion focus CEOs, the likelihood of breadth search increases faster, i.e., the slope is greater, when compared to the high promotion focus CEOs" (lines 424-429). Unfortunately, this is not clear from Figure 1.

How was the impact of positive performance feedback on innovation search presented in Table 3 calculated (by what method)?

Round 2

Reviewer 3 Report

I would like to thank the authors for the effort put into improving the article in accordance with the comments of the reviewer. I believe the article is now much clearer to the reader. I fully accept all changes proposed by the authors.